# Pan-Mitogenome Construction, Intraspecific Variation, and Adaptive Evolution of the Plant Pathogenic Fungus *Claviceps purpurea*

**DOI:** 10.3390/biology14111548

**Published:** 2025-11-04

**Authors:** Mingliang Ding, Rui Hu, Jinlong Jia, Cuiyuan Wei, Yongzhen Cui, Hefa Liao, Zhuliang Yang, Jianwei Guo, Zhanhong Ma, Yuanbing Wang

**Affiliations:** 1Department of Plant Pathology, College of Plant Protection, China Agricultural University, Beijing 100193, China; dml@yaas.org.cn; 2Food Crops Research Institute, Yunnan Academy of Agricultural Sciences, Kunming 650205, China; cyz@yaas.org.cn; 3State Key Laboratory of Phytochemistry and Natural Medicines, Kunming Institute of Botany, Chinese Academy of Sciences, Kunming 650201, China; 18187890265@163.com (R.H.); 15893482367@163.com (J.J.); wcy1372812954@163.com (C.W.); liao18788020747@163.com (H.L.); 4Yunnan Key Laboratory for Fungal Diversity and Green Development, Kunming Institute of Botany, Chinese Academy of Sciences, Kunming 650201, China; fungi@mail.kib.ac.cn; 5College of Traditional Chinese Medicine, Yunnan University of Chinese Medicine, Kunming 650500, China; 6Institute of International Rivers and Eco-Security, Yunnan University, Kunming 650504, China; 7College of Agronomy and Life Sciences, Yunnan Urban Agricultural Engineering and Technological Research Center, Kunming University, Kunming 650214, China; gjwkf475001@sina.com

**Keywords:** *Claviceps purpurea*, mitochondrial genome, phylogeny, protein-coding genes

## Abstract

**Simple Summary:**

Fungi of the genus *Claviceps* produce ergot alkaloids, a group of compounds with both 23 toxic and medicinal properties. Despite its significance in agriculture and medicine, little 24 is known about its mitochondrial genome. This study assembled, annotated, and analyzed 25 the mitochondrial genomes of 15 different strains from diverse host plants and regions. 26 Analysis of the *C. purpurea* pan-mitogenome demonstrated that the accessory regions, 27 with an average proportion of 48.23%, are the main contributor to mitogenome variation. 28 Analysis of the 14 protein-coding genes revealed Ka/Ks ratios below 1, indicating strong 29 purifying selection. Notably, the *atp9* gene was absent in all strains, suggesting a potential 30 adaptive gene loss. Structural variations were predominantly located in the intergenic region 31 between *rns* and *rnl*. Phylogenetic analysis based on concatenated mitochondrial 32 genes placed *Claviceps* as most closely related to the genus *Epichloë*. The 15 *C. purpurea* 33 strains grouped into five well-supported subclades, with Chinese and non-Chinese isolates 34 forming distinct lineages. Among these, the Chinese strains ACCC 37001 and 35 KUNCC 11030 represented the earliest diverging lineages. This research helps us better 36 understand how this pathogen evolves and adapts to different environments, which may 37 aid in future efforts to control crop diseases and utilize its beneficial properties.

**Abstract:**

*Claviceps purpurea* is a specialized phytopathogenic fungus that infects grasses and produces pharmacologically active compounds, attracting considerable interest in genomic research. In this study, we assembled and annotated the complete mitogenomes of 15 *C. purpurea* strains isolated from different host plants, including seven newly sequenced isolates from China. Analysis of the *C. purpurea* pan-mitogenome demonstrated that the accessory regions, with an average proportion of 48.23%, are the main contributor to mitogenome variation. Analysis of the 14 protein-coding genes revealed Ka/Ks ratios below 1, indicating strong purifying selection. Notably, the *atp9* gene was absent in all strains, suggesting a potential adaptive gene loss. Structural variations were predominantly located in the intergenic region between *rns* and *rnl*. Phylogenetic analysis based on concatenated mitochondrial genes placed *Claviceps* as most closely related to the genus *Epichloë*. The 15 *C. purpurea* strains grouped into five well-supported subclades, with Chinese and non-Chinese isolates forming distinct lineages. Among these, the Chinese strains ACCC 37001 and KUNCC 11030 represented the earliest diverging lineages. This study elucidates the intraspecific variation and evolutionary patterns of the mitogenome in *C. purpurea* and highlights the value of mitogenome in resolving phylogenetic relationships.

## 1. Introduction

*Claviceps purpurea*, the type species of the genus *Claviceps* (Clavicipitaceae, Hypocreales), is a phytopathogenic fungus that specifically infects the ovaries of grass species (Poaceae) [1]. It exhibits a remarkably broad host range, parasitizing more than 400 plant species spanning three subfamilies and 19 genera within the family Poaceae, including economically critical cereal crops such as barley, wheat, and rye [2]. With worldwide distribution, *C. purpurea* poses substantial agricultural threats across diverse ecosystems [1]. Beyond its role as a plant pathogen, *C. purpurea* is renowned for producing toxic ergot alkaloids within its sclerotia. These metabolites can contaminate cereal grains, and their inadvertent consumption may result in severe human and animal health disorders referred to as ergotism [3,4,5]. Paradoxically, ergot alkaloids also possess valuable pharmacological properties and have been employed therapeutically in the prevention of postpartum hemorrhage [6], treatment of migraines [7], Parkinson’s disease [8], type II diabetes [9], prolactinomas [10], Alzheimer’s disease [11], and several psychiatric disorders [12]. This dual nature as both a dangerous pathogen and a pharmaceutical resource makes *C. purpurea* and its metabolites a subject of sustained scientific interest, with research spanning agriculture, ecology, medicine, and biotechnology.

The mitochondrial genome (mitogenome), often termed the “second genome” in eukaryotes, offers several advantages as a molecular marker [13,14,15]. Its maternal inheritance pattern eliminates the complications associated with biparental inheritance and frequent recombination in nuclear DNA, enabling clearer tracing of maternal lineages [14]. A generally low recombination rate also contributes to structural stability of the genome, which aids in reliable phylogenetic inference [15]. Moreover, due to the relatively conserved evolutionary rate of the mitogenome, mitochondrial DNA allows for the selection of genetic markers that are applicable at different taxonomic levels, including the differentiation of closely related species as well as the exploration of broader evolutionary patterns and species origins [16]. These characteristics make the mitogenome an indispensable tool in the fields of evolutionary biology and phylogenetics [17].

In fungi, the mitogenome typically consists of a conserved set of protein-coding genes (PCGs) involved in oxidative phosphorylation and electron transport, two ribosomal RNA (rRNA) genes, a variable number of transfer RNA (tRNA) genes, and several unidentified open reading frames (uORFs). Together, these elements govern mitochondrial function and gene expression, further enhancing the genomic utility as a molecular marker [18,19]. However, mitogenome data for species within the genus *Claviceps* remain strikingly limited. The NCBI database currently contains only a handful of records for mitogenomes within this genus, severely limiting comprehensive research. To date, only one strain, *C. purpurea* 20.1, has had its mitogenome fully sequenced and described. This genome spans 55,537 base pairs, exhibits a high AT content (65%), and exists as a closed circular double-stranded molecule comprising 31 genes, including 14 PCGs, 2 rRNAs, and 16 tRNAs [20]. This limited dataset reflects the early stage of mitogenome research in *Claviceps*, highlighting the urgent need for further exploration. The lack of data on genomic diversity, evolutionary dynamics, and functional roles calls for more in-depth studies to unlock the full potential of mitogenomes as molecular markers. Expanding this knowledge could significantly enhance taxonomy, phylogeny, and evolutionary biology of *Claviceps* species.

Given the multifaceted importance of *C. purpurea* in agriculture, ecology, and medicine, along with the notable scarcity of mitogenomic data, this study aims to address a critical knowledge gap. A key strength of this research lies in its diverse dataset, comprising 15 *C. purpurea* strains collected from the host plants Roegeria kamoji, Avena sativa (oat) and Bromus riparius in China, Canada and the United Kingdom. These strains were isolated from various host plants, such as *Roegeria kamoji*, *Avena sativa* (oat), and *Bromus riparius*, reflecting both ecological and host diversity. Through a comprehensive analysis of the mitogenomes of these strains, this study aims to elucidate the intraspecific genetic diversity of *C. purpurea*, characterize the structural and evolutionary characteristics of its mitogenome, and explore potential mechanisms underlying host adaptation.

## 2. Materials and Methods

### 2.1. Sample Collection

The 15 *C. purpurea* strains from host plants belonging to 13 genera and collected in three distinct regions (China 7, Canada 7 and the United Kingdom 1) were evaluated. However, Chinese strains account for a high proportion. This may limit the global applicability of the findings to some extent. Future studies will incorporate strains from more geographic regions worldwide to comprehensively reveal the genetic diversity and evolutionary history of *C. purpure*. Among these, seven strains (ACCC 36989, ACCC 37001, ACCC 37007, ACCC 37016, KUNCC 10878, KUNCC 11030, and KUNCC 11035) were sequenced in this study using second-generation sequencing technologies. The remaining eight strains (LM33, LM46, LM60, LM71, LM207, LM232, LM223, and LM474) were sourced from the NCBI Sequence Read Archive (SRX5575019, SRX5575017, SRX5575016, SRX5575011, SRX5575010, SRX5574998, SRX5574999, and SRX5574993; available online at: https://www.ncbi.nlm.nih.gov/sra/, accessed on 11 December 2024). Detailed information on host species, geographic origin, and collection metadata for each strain is provided in Appendix A.

### 2.2. Mitogenome Sequencing, Assembly, and Annotation

The mitogenomes of *C. purpurea* strains were sequenced by BGI Genomics Co., Ltd. (Wuhan, China) using the DNBSEQ-T7 sequencing platform. Genome assembly was performed using GetOrganelle v1.7.7.0 [21] and NOVOPlasty v4.3.1 [22] via de novo assembly, resulting in complete mitogenomes for all 15 *C. purpurea* strains. Among these, GetOrganelle recruits organelle-associated reads using a modified ‘baiting and iterative mapping’ approach, conducts de novo assembly, filters and disentangles the assembly graph, and produces all possible configurations of circular organelle genomes. Genome annotation was conducted using the online tools MFannot (available online at: https://megasun.bch.umontreal.ca/cgi-bin/mfannot/mfannotInterface.pl/, accessed on 31 December 2024) and MITOS2 [23], based on mitochondrial genetic code 4. MFannot effectively captures evolutionarily conserved feature signals in protein primary sequences and secondary structures by integrating Hidden Markov Models (HMMs), Covariance Models (CMs), and the ERPIN model. Transfer RNA (tRNA) genes and their secondary structures were identified using tRNAscan-SE v1.23 [24]. Circular genome maps for each strain were generated using the OGDraw v1.3.1 tool [25].

### 2.3. Comparative Analysis of Mitogenome

The basic features of the mitogenomes from 15 *C. purpurea* strains were analyzed using PhyloSuite v1.2.3 [26,27]. Nucleotide compositional biases were quantified using the following formulas: GC skew = (G − C)/(G + C) and AT skew = (A − T)/(A + T). Relative synonymous codon usage (RSCU) [28] was calculated with CodonW v1.4.2 (available online at: http://codonw.sourceforge.net/, accessed on 10 January 2025). Synonymous (Ks) and nonsynonymous (Ka) substitution rates for 14 PCGs were estimated using KaKs_Calculator v2.0 [23,29]. Genetic distances among strains were computed using MEGA v11.0 [30] based on the Kimura 2-parameter (K2P) model [31]. Nucleotide diversity (Pi) and highly variable regions were identified using DnaSP v6.12.0 [32], with a sliding window of 400 bp and a step size of 200 bp. For a comprehensive evaluation of the structure of repeat elements, simple sequence repeats (SSRs) were identified using the MISA web platform [33], tandem repeats were detected with the Tandem Repeats Finder (TRF) tool [34], and dispersed repeats were analyzed using the REPuter server [35].

### 2.4. Phylogenetic Analysis

To explore the phylogenetic relationships among *C. purpurea* strains from different host plants, mitogenomes of 11 additional species within the family Clavicipitaceae were retrieved from the NCBI database (Appendix A). Two species from the family Cordycipitaceae, *Cordyceps cicadae* (MH92223) and *Cor. tenuipes* (NC001329), were designated as the outgroup to determine the phylogenetic placement of *C. purpurea*. The PCGs from all taxa were extracted using PhyloSuite v1.2.3 [26]. Multiple sequence alignments were conducted with MAFFT v2.1 [36], and poorly aligned regions were trimmed using trimAl v1.4.rev15 [37]. The resulting alignments were concatenated into a supermatrix using PhyloSuite [38,39,40]. Maximum likelihood analysis was performed using IQ-TREE v2.1.3 [41], and Bayesian inference analysis was conducted using MrBayes v3.2 [42]. The final phylogenetic tree was visualized with FigTree v1.4.4 (available online at: http://tree.bio.ed.ac.uk/software/figtree/, accessed on 15 January 2025).

### 2.5. Pan-Mitogenome Analysis

The core mitogenome of each *C. purpurea* strain was identified using the bioinformatics tools Spine [43] and AGEnt [44,45], enabling the distinction between conserved and accessory genomic elements. To further investigate the accessory genomic elements (AGEs), their nucleotide sequences were clustered using ClusTAGE, allowing for the identification of a minimal set of AGEs shared among the strains and the determination of their distribution across the mitogenomes. Finally, the spatial organization of these clustered elements was visualized using ClusTAGE Plot, which generated circular diagrams illustrating the genomic locations of AGEs in each strain.

## 3. Results

### 3.1. Mitogenome Characteristics

In this study, we assembled and annotated the mitogenomes of 15 *C. purpurea* strains collected from diverse geographic regions and host plants. All mitogenomes were circular DNA molecules ranging in size from 42,267 to 51,342 bp (Figure 1), with the strains LM46 and KUNCC 11035 exhibiting the smallest and the largest genomes, respectively. The GC content across these genomes varied from 34.7% to 35.2%, with an average of 35.0%, reflecting a strong A/T base composition bias. GC-skew values ranged from 0.07 to 0.091, indicating a moderate asymmetry in base composition. Gene annotation revealed that each strain contained 42 to 48 genes (Appendix A). All strains contained 14 core protein-coding genes (PCGs), typical of fungal mitogenomes, including seven NADH dehydrogenase subunits (*nad1*–*nad6*, *nad4L*), three cytochrome c oxidase subunits (*cox1*, *cox2*, *cox3*), two ATP synthase subunits (*atp6*, *atp8*), and one cytochrome b gene (*cob*). Notably, the *atp9* gene was absent in all strains. In addition to PCGs, each genome contained two ribosomal RNA genes (*rnl* and *rns*), the ribosomal protein S3 gene (*rps3*) embedded within *rnl*, and 25 tRNA genes representing all 20 standard amino acids. In each *C*. *purpurea* strain, one to seven open reading frames (ORFs) were detected within the intergenic regions. This variation in ORF presence and quantity among strains was a key factor contributing to the intraspecific diversity of mitogenome composition. Analysis of the mitogenomes from 15 *C. purpurea* strains isolated from different host species revealed that exon numbers ranged from 28 to 36, with intron numbers varying from 20 to 30 PCGs accounting for 24.72% to 35.63% of the total mitogenome length, showing the extensive variation in gene composition within *C. purpurea*.

In the mitogenomes of the 15 *C. purpurea* strains, intergenic and intronic regions collectively accounted for more than 64% of the total genome length. In contrast, rRNA and tRNA genes comprised only 10.46% and 3.91% of the genome, respectively (Figure 2). Correlation analysis revealed strong positive associations between mitogenome size and the total lengths of ORFs (r = 0.5604), intergenic regions (r = 0.5388), and intronic regions (r = 0.3449). These results indicated that variations in the intergenic region and intron region were the primary drivers of mitogenome size differences in *C. purpurea*.

### 3.2. Codon Usage

To further investigate the conservation and variation of mitogenome PCGs among the 15 *C. purpurea* strains, codon usage patterns were analyzed (Figure 3). The results showed that all other 14 PCGs across the genomes consistently used ATG as the start codon, with the exception of the *atp6* that initiated with TTG. In terms of stop codons, *atp6* and *cytb* terminated with ATG, whereas *atp8*, *cox2*, *cox3*, *nad4*, *nad4L*, *nad6*, and *rps3* terminated with TAA. For *cox1*, *nad1*, *nad2*, and *nad3*, both ATG and TAA were identified as stop codons across different strains (Appendix A).

Codon usage analysis revealed a high degree of similarity in codon frequency across strains isolated from different host plants. Overall, codons ending in A/T were used more frequently than those ending in G/T, indicating a strong AT bias. The most frequently used codons were UAA (coding for leucine) and AGA (coding for arginine). The elevated AT content of the mitogenomes (averaging 65.04%) was primarily attributable to the preferential use of codons ending in adenine (A) or thymine (T) (Appendix A). Although strains from different host sources exhibited high overall similarity in codon usage frequency (generally favoring A/T-terminating codons), they formed three distinct clusters based on RUSC profiles. This indicates that codon preferences are not entirely uniform among strains, with LM207 showing significant differences in codon usage preferences compared to other strains.

### 3.3. Variation in Mitogenome PCGs

Among the 14 mitogenome PCGs, *cox1*, *nad1*, and *nad3* exhibited notable variation in sequence length across *C. purpurea* strains isolated from different host plants. The most substantial variation was observed in *nad3*, with sequence lengths ranging from 363 bp to 1662 bp. In contrast, *cox2*, *nad4*, *nad4L*, and *nad5* were highly conserved in length (Figure 4A). With respect to GC content, *nad1* showed the greatest variability, ranging from 35.2% to 39.0%. The *cox2* gene exhibited the highest average GC content (39.5%), whereas *rps3* had the lowest (30.4%) (Figure 4B). Nucleotide skew analysis revealed that all genes except *atp6*, *rps3*, and certain variants of *nad4* displayed negative AT skew (Figure 4C), indicating a preference for thymine over adenine. In terms of GC skew, most core PCGs showed positive values, with the exception of *atp8*, *cytb*, and some *cox1* variants, which exhibited negative skew (Figure 4D). Collectively, these results showed substantial variability in the base composition of mitogenome PCGs in *C. purpurea*, reflecting diverse evolutionary pressures acting on different genes (Appendix A).

Selection pressure analysis was conducted on 14 PCGs from 15 *C. purpurea* strains isolated from different host plants (Figure 5), suggesting that the substitution rates varied across genes. The nonsynonymous substitution rate (Ka) was highest in *cox3* and *nad1*, while the lowest Ka values were observed in *atp8* and *nad4L*. In terms of synonymous substitution rates (Ks), *cox1* and *cox2* showed the highest values, whereas *atp8* and *nad6* exhibited the lowest. Importantly, all PCGs had Ka/Ks ratios less than 1, indicating that they were subject to purifying selection. This suggested that their amino acid sequences were relatively conserved during evolution, with *atp8* and *nad4L* showing particularly strong selective constraints. Genetic distance analysis further supported these results: the *cox2* gene exhibited the greatest genetic distance, indicating a higher evolutionary rate and greater variability among strains. In contrast, *atp8* showed the smallest genetic distance, making it the most conserved gene across the mitogenomes of *C. purpurea*.

### 3.4. Repeat Sequence

Analysis of repeat sequences in the mitogenomes of 15 *C. purpurea* strains from diverse host plants showed considerable variability in repeat content and distribution (Figure 6). Each genome contained between 3 and 14 simple sequence repeat (SSR) regions, ranging from 35 to 225 bp in length (Appendix A). Mononucleotide and trinucleotide repeats were consistently present across all strains, whereas pentanucleotide and hexanucleotide repeats appeared only in a subset of genomes. No dinucleotide or tetranucleotide repeats were detected in any strain. Tandem repeat analysis identified 2 to 6 repeat regions per genome, with repeat lengths ranging from 24 to 55 bp (Appendix A). The strain KUNCC 11030 exhibited the highest number of tandem repeats, whereas strains LM33 and LM46 had the fewest. Notably, the six strains originating from China generally possessed shorter tandem repeat regions. Dispersed repeat analysis showed that forward repeats (F) were the most abundant, while complementary repeats (C) were observed only in a few strains (Appendix A). These results suggest that mitochondrial repeat sequences in *C. purpurea* vary significantly among strains from different host origins, potentially reflecting underlying genomic plasticity or evolutionary divergence.

### 3.5. Nucleotide Diversity

Nucleotide diversity (Pi) of mitogenome PCGs across 15 *C. purpurea* strains was assessed using a sliding window analysis (Figure 7). The Pi values ranged from 0.002 to 0.023. Notably, the regions corresponding to *orf139* and *orf143* exhibited significantly higher nucleotide diversity than other genomic regions, indicating elevated levels of genetic variation in these loci among strains from different hosts. These results suggested that *orf139* and *orf143* might represent genomic hotspots of variation and have potential as molecular markers for distinguishing *C. purpurea* strains adapted to different host species.

### 3.6. Intron Distribution

Intron distribution analysis of the mitogenomes of *C. purpurea* identified a total of 380 introns, including 22 Group II and 134 Group I introns (Figure 8). The number of introns per strain ranged from 20 to 30 and were distributed across eight genes: *rnl*, *cox2*, *nad5*, *cob*, *cox1*, *nad1*, *atp6*, and *cox3*. Among these, atp6 exhibited a conserved number of introns across all strains, whereas cob and cox1 showed frequent intron gain or loss events. Strains LM46, LM474, LM71, and LM60, forming part of the earliest diverging lineage, exhibited intron loss in the *rnl* gene. Additionally, only strain LM207 acquired a Group IA intron in the *cob* gene. In contrast, the genes *atp8*, *nad2*, *nad3*, *nad4*, *nad4L*, *nad6*, and *rps3* were intronless in all 15 strains examined. These results indicated that intron gain and loss occurred during the evolutionary adaptation of *C. purpurea* to different host plants, contributing to intraspecific genomic differentiation.

### 3.7. Structural Variation

Extensive structural variations were identified in the mitogenomes of the 15 *C. purpurea* strains, primarily involving non-core genes. In contrast, the core genes exhibited a conserved gene count and order across all strains. The structural variations associated with non-core regions were mainly concentrated in two specific genomic intervals. The first region of variation lies between the *rnl* and *nad2* genes, where insertion and deletion events of *orf* genes were frequently detected. The second, more variable region spanned from *rns* to *rnl*. Except for strains KUNCC 11035, LM71, LM474, and LM46, all other strains exhibited one or more reverse-oriented *orf* insertions in this region. Moreover, insertions of the *dpo* gene were detected in strains ACCC 37016, KUNCC 10878, LM232, LM207, and LM60, with insertion sites varying across strains (Figure 9). These results highlighted the dynamic nature of non-core genomic regions of *C. purpurea*.

### 3.8. Pan-Mitogenome

A pan-mitogenome was constructed based on 15 *C. purpurea* strains. With respect to gene composition, the core genome consisted of PCGs, rRNA genes, and tRNA genes, all of which were completely conserved across strains. These core elements were primarily involved in essential biological processes, such as electron transport and biosynthesis. In contrast, gene-associated introns, *orf* genes, and repeat sequences were classified as components of the accessory genome or regions. These elements might contribute to the adaptive evolution of *C. purpurea* [46]. At the sequence level, core genomic regions accounted for 46.99% to 57.91% of the mitogenome, with an average of 51.76%. The smallest proportion of core sequences was found in the strain ACCC 36989 (46.99%), whereas the largest was observed in the strain LM46 (57.91%). Accessory sequences comprised a substantial portion of the genome, ranging from 42.07% to 52.97%, with an average of 48.23%. The strain ACCC 36989 exhibited the highest proportion of accessory sequences (Figure 10). A significant correlation was found between the abundance of accessory sequences and overall mitogenome size, showing that these variable regions are the primary drivers of genome size variation in *C. purpurea*.

### 3.9. Phylogenetic Analysis of the Clavicipitaceae

A phylogenetic tree the family Clavicipitaceae was reconstructed using a supermatrix composed of concatenated sequences from 14 PCGs, and the phylogenetic analysis was conducted employing both Maximum likelihood and Bayesian inference methods, resulting in a well-resolved phylogenetic tree (Figure 11). The results supported the division of the 26 species into five major clades, which correspond to the genera *Claviceps*, *Epichloë*, *Pochonia*, *Metarhizium*, and *Orbiocrella*. Of particular note, *Claviceps* was found to be the closest to *Epichloë*. All 15 *C. purpurea* strains included in this study clustered within the *Claviceps* clade and were further resolved into five distinct subclades. Geographically, strains originating from China formed separate sublineages compared to those from other regions. Notably, two Chinese strains (ACCC 37001 and KUNCC 11030) formed the earliest diverging lineage with the basal phylogenetic position of these isolates, suggesting a potential East Asian origin for early evolutionary lineages of *C. purpurea*.

## 4. Discussion

### 4.1. Structural Characteristics of the C. purpurea Mitogenome

Fungal mitogenomes have undergone extensive gene reduction throughout evolution, retaining only a core set of protein-coding and ribosomal RNA genes essential for respiration and protein synthesis [47,48,49]. However, they still harbor variable numbers of introns, ORFs, repetitive elements, and plasmid-derived DNA fragments, which contribute to the structural complexity and diversity of fungal mitogenomes [19,50]. In this study, we assembled and annotated the mitogenomes of 15 *C. purpurea* strains isolated from diverse host plants and geographic regions. All assemblies revealed circular DNA molecules, ranging from 42,267 to 51,342 bp in length, with an average size of 47,870 bp. The average GC content was 65.04%, showing a pronounced AT bias. Gene content varied from 42 to 48 genes per strain, consistently 14 PCGs (with universal absence of *atp9*), 2 rRNA genes, 25 tRNA genes, and several strain-specific ORFs. Most core PCGs used the standard start codon (ATG) and stop codon (TAA). These findings reveal considerable intraspecific variation in genome size, GC content, and nucleotide composition across *C. purpurea* strains from different hosts and regions. Such genomic variation may affect gene expression and protein function, potentially influencing host-specific adaptation and evolutionary divergence. Genome size divergence is strongly correlated with expansions or contractions in intergenic spacers and accessory ORFs, consistent with previous reports highlighting the role of non-coding regions in fungal mitogenome evolution [28,29,30]. Detailed analysis of the gene components showed that most intraspecific differences in *C. purpurea* mitogenomes are concentrated in intergenic and intronic regions. Specifically, variation in genome size is primarily driven by the expansion and contraction of intergenic sequences and ORF regions.

Despite notable size variation, the order and content of core PCGs, rRNAs, and tRNAs are highly conserved across strains, likely reflecting functional constraints related to energy production and cellular viability. A notable finding was the consistent absence of *atp9*, which encodes a subunit of the ATP synthase complex—typically considered a core mitochondrial gene [51]. This loss is unusual, as *atp9* is retained in other genera of Clavicipitaceae such as *Orbiocrella* [29], *Pochonia* [52], *Epichloë* [53], and *Metarhizium* [54]. A similar loss has been documented in *Corynespora cassiicola*, the causal agent of rubber tree target spot disease [50], suggesting that *atp9* loss in *C. purpurea* may reflect lineage-specific adaptation to its mitochondrial environment [55]. This level of gene loss raises an important question: to what extent can the function of *atp9* be preserved or compensated for in its absence, and how does the ATP synthase complex maintain functionality without this subunit?

### 4.2. Intraspecific Variation and Evolutionary Dynamics

Although mitochondrial PCGs in *C. purpurea* are generally conserved, evolutionary divergence has occurred alongside host range expansion, leading to detectable variability at the genomic level [56,57]. Our analysis revealed substantial intraspecific variation among the strains studied here, including variation in PCG count (42–48 genes), largely attributable to differences in accessory *orf* content. Introns are also highly dynamic, ranging from 20 to 30 per genome. These introns are distributed across eight genes, *rnl*, *cox2*, *nad5*, *cob*, *cox1*, *nad1*, *atp6*, and *cox3*, with the highest mobility observed in *cob* and *cox1*, consistent with previous studies [58,59,60]. Repetitive sequences are also abundant in these genomes, with dispersed repeats being the most common, classified as forward, reverse, complementary, and palindromic types. Collectively, *orf* dynamics, intron variation, and repeat sequence expansion contribute to mitogenome size variation and likely play a key role in host adaptation and evolutionary processes in *C. purpurea*. Pan-mitogenome analysis further revealed that the core genomic regions account for approximately 51.76% of the genome, whereas variable regions comprise about 48.24%, underscoring the extent of genetic divergence within the species.

Genetic diversity analysis revealed high-nucleotide-diversity (Pi) regions in the mitogenomes of 15 *C. purpurea* strains. These high-variability loci are primarily located in non-coding regions, such as *orf139* and *orf143*, highlighting their potential as candidate regions for the development of DNA barcodes [61]. Previous research indicates that *orf139* and *orf143* are two common open reading frames in mitochondrial genomes. They are factors associated with cytoplasmic male sterility (CMS) in plant organelle genomes, particularly mitochondria [62,63]. Therefore, these genes with high nucleotide diversity may exert significant effects on their functions in mitochondrial physiology and reproductive development. Moreover, all core PCGs exhibit Ka/Ks ratios below 1, indicating that they are evolving under purifying selection. Among these, *nad6* displayed the highest evolutionary rate, suggesting it may be subject to more relaxed selective constraints or functional divergence.

Structural variation in mitochondrial genes provides valuable information for understanding phylogenetic relationships and evolutionary dynamics in eukaryotes [64,65,66,67,68]. In this study, extensive structural variation was identified in the mitogenomes of *C. purpurea*, particularly concentrated in two genomic regions: (1) the region between *rnl* and *nad2*, where insertion and deletion events involving *orf* genes were common; and (2) the region spanning *rns* to *rnl*, which include rare insertions of the *dpo* gene. Previous studies have indicated that the accumulation of repetitive elements is closely linked to structural rearrangements in fungal mitogenomes, supporting the hypothesis that such structural plasticity may play a role in species adaptation and genome evolution [69,70,71].

### 4.3. Molecular Phylogeny and Biogeographic Inference

Mitogenomes, characterized by their compact size, rich genetic diversity, conserved PCGs, and maternal inheritance, have proven to be powerful tools for studies in population genetics, phylogenetics, and evolutionary biology [72,73,74,75]. In this study, a maximum likelihood phylogenetic tree of Clavicipitaceae was constructed using a supermatrix of 14 concatenated mitochondrial PCGs from 26 taxa representing five genera, with two taxa of Cordycipitaceae as the outgroup. The resulting phylogeny resolved these fungi into five distinct clades, corresponding to *Claviceps*, *Epichloë*, *Pochonia*, *Metarhizium*, and *Orbiocrella*, with *Claviceps* forming the closest relationship to *Epichloë*. This topology is consistent with the multi-gene phylogeny conducted by previous [76]. As a specialized pathogen that infects the ovaries of grasses, the evolutionary trajectory of *C. purpurea* is strongly linked to its host adaptation. Our phylogenetic analysis revealed *C. purpurea* strains from different host plants formed five distinct subclades, likely reflecting divergence associated with host specialization. Most strains clustered by geographic origin, with clear separation between Chinese and non-Chinese isolates. Notably, the eight *C. purpurea* strains from Canada and UK are distributed across three clades, consistent with previous population genetic studies that identified multiple subpopulations in North America. By incorporating a broader sampling of Chinese strains from diverse hosts, our study uncovered a more complex phylogenetic structure comprising five subclades. Strikingly, the Chinese strains ACCC 37001 and KUNCC 11030 formed the earliest-diverging lineage, suggesting that they may represent ancestral types within the species. This finding supports the hypothesis that East Asia—and specifically China—may have been a center of early diversification for this species.

## 5. Conclusions

This study provides the first in-depth pan-mitogenomic perspective of the phytopathogenic fungus *C. purpurea*. It is revealed that the mitogenome is circular, AT-rich, and contains a highly conserved core gene set, but lacks the *atp9* gene across all examined strains. Genome size variation was found to be largely attributable to the dynamic expansion and contraction of accessory genomic elements, particularly introns, ORFs, and intergenic repeats, which comprise nearly half of the mitogenome and serve as a major source of intraspecific diversity. Substantial intraspecific variation was found in GC content, coding sequence lengths, intron number and distribution, as well as the composition and abundance of repetitive elements. Despite these variations, all PCGs were found to be under purifying selection, indicating evolutionary constraints on their function. Phylogenomic reconstruction strongly supported a sister-group relationship between *Claviceps* and *Epichloë*, and identified five well-supported subclades within *C. purpurea*, with phylogeny largely correlating with geographic origin. The basal position of Chinese strains suggests East Asia as a potential center of early diversification for this species. The structural plasticity of the mitogenome, primarily driven by variation in accessory regions, likely plays a crucial role in host adaptation and ecological divergence. These findings substantially advance our understanding of the mitogenomic characteristics, evolutionary history, and host-associated genetic divergence of this significant pathogen.

## Figures and Tables

**Figure 1 biology-14-01548-f001:**
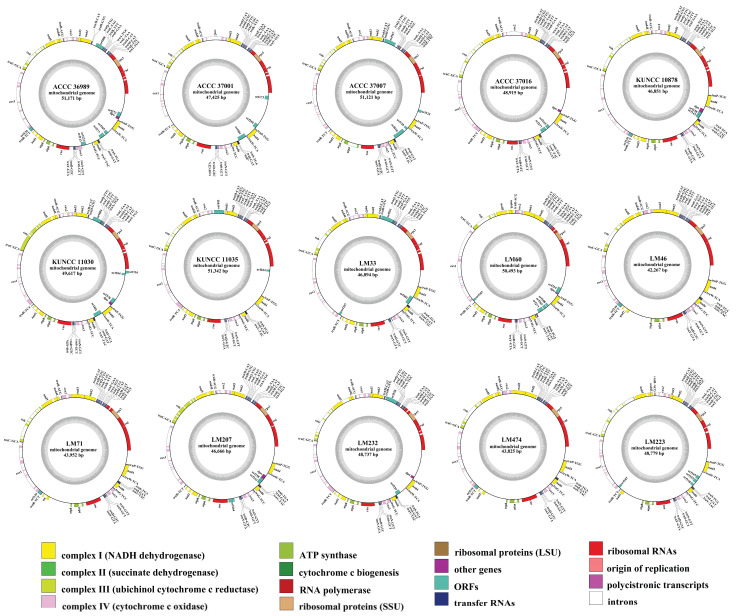
Annotated circular maps of 15 *Claviceps purpurea* mitogenomes from different host plants. The genes with distinct functions are represented by different colored blocks. Genes located on the negative strand are shown inside the circle, while genes on the positive strand are displayed outside the circle.

**Figure 2 biology-14-01548-f002:**
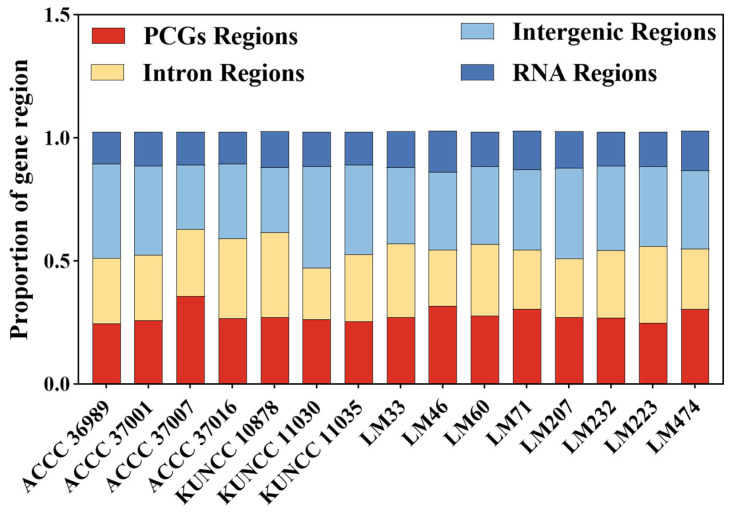
Relative proportions of different gene regions in the mitogenomes of 15 *Claviceps purpurea* strains. Red block for protein-coding genes (PCGs); yellow block for intron regions; light blue block for intergenic regions; and dark blue block for RNA regions (rRNA + tRNA).

**Figure 3 biology-14-01548-f003:**
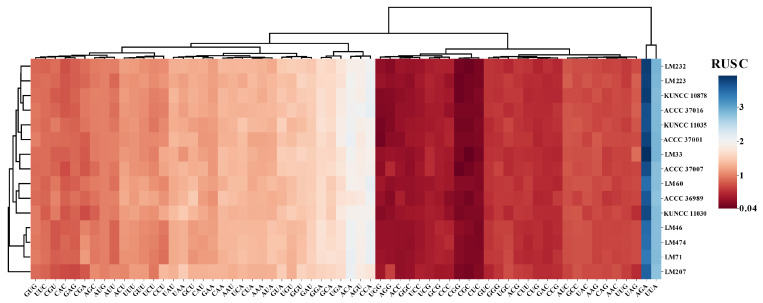
Relative synonymous codon usage (RSCU) in the mitogenomes of 15 *Claviceps purpurea* strains. The *x*-axis represents codons, the *y*-axis corresponds to *C. purpurea* strains, with color intensity indicating RSCU values (blue: higher RSCU; red: lower RSCU).

**Figure 4 biology-14-01548-f004:**
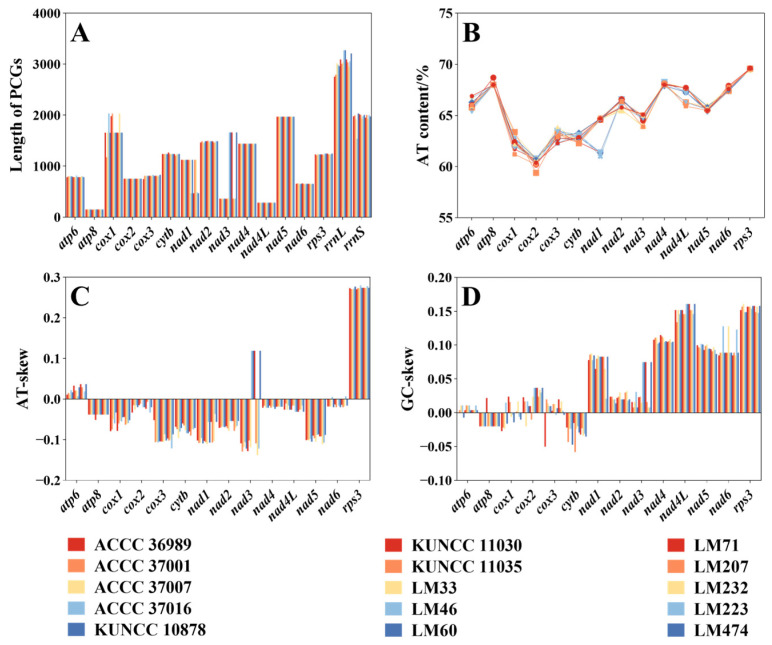
Sequence variation of protein-coding genes (PCGs) in the mitogenomes of 15 *Claviceps purpurea* strains. (**A**) Variation in the length of 14 PCGs; (**B**) Variation in AT content; (**C**) Variation in AT-skew; (**D**) Variation in GC-skew.

**Figure 5 biology-14-01548-f005:**
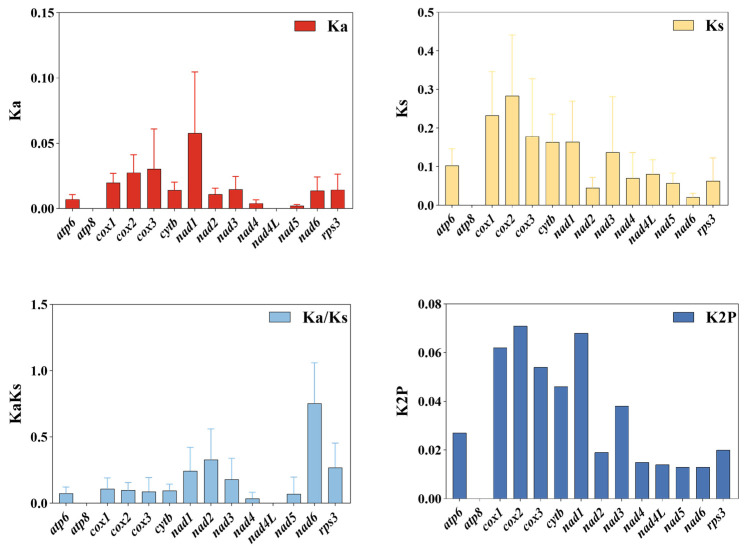
Genetic distance and evolutionary rate of core protein-coding genes (PCGs) in the mitogenomes of 15 *Claviceps purpurea* strains.

**Figure 6 biology-14-01548-f006:**
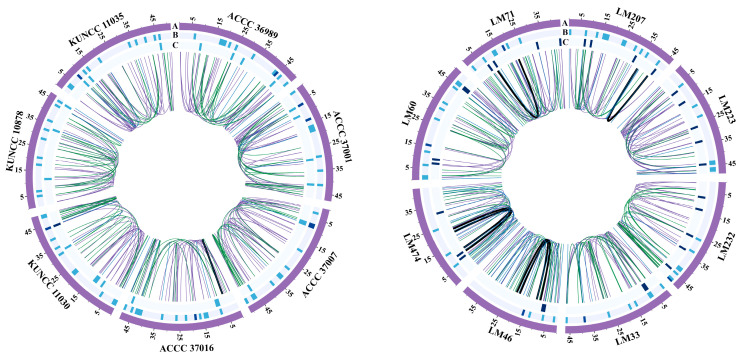
Mitogenome repeat sequence of 15 *Claviceps purpurea* strains. The circles in the diagram present the positions of dispersed repeats, tandem repeats, and simple repeats from innermost to outermost. The blue-gray lines indicate forward repeats, and green lines represent palindromic repeats.

**Figure 7 biology-14-01548-f007:**
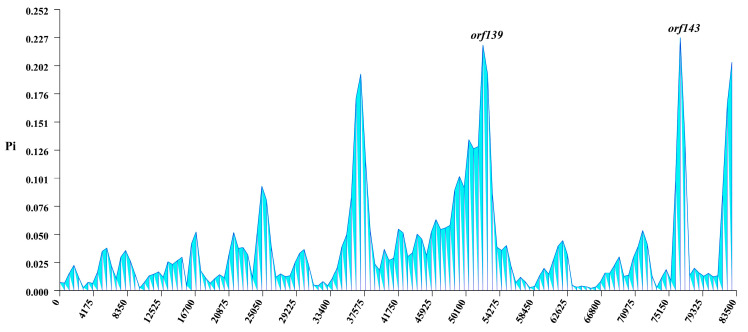
Nucleotide diversity of mitogenomes of 15 *Claviceps purpurea* strains. The *x*-axis represents mitogenome positions, and the *y*-axis indicates nucleotide diversity (Pi) values.

**Figure 8 biology-14-01548-f008:**
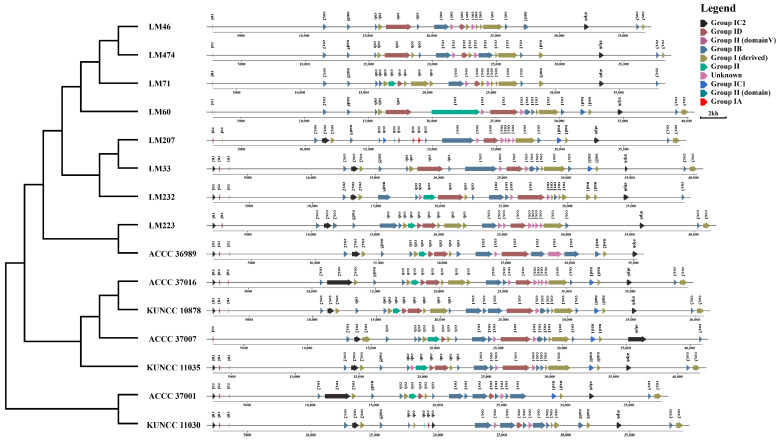
Distribution of introns in the mitogenomes of 15 *Claviceps purpurea* strains. The (**left**) panel displays the maximum likelihood phylogenetic tree constructed based on core PCGs, and the (**right**) panel shows the positional information of introns in the mitochondrial genome.

**Figure 9 biology-14-01548-f009:**
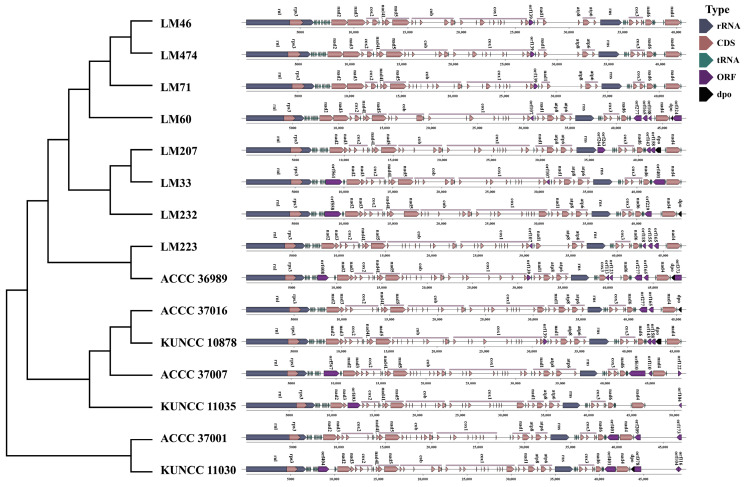
Structural variation in the mitogenomes of 15 *Claviceps purpurea* strains. The (**left**) panel shows the maximum likelihood phylogenetic tree constructed based on core PCGs, and the (**right**) panel displays the positional information of mitochondrial genes.

**Figure 10 biology-14-01548-f010:**
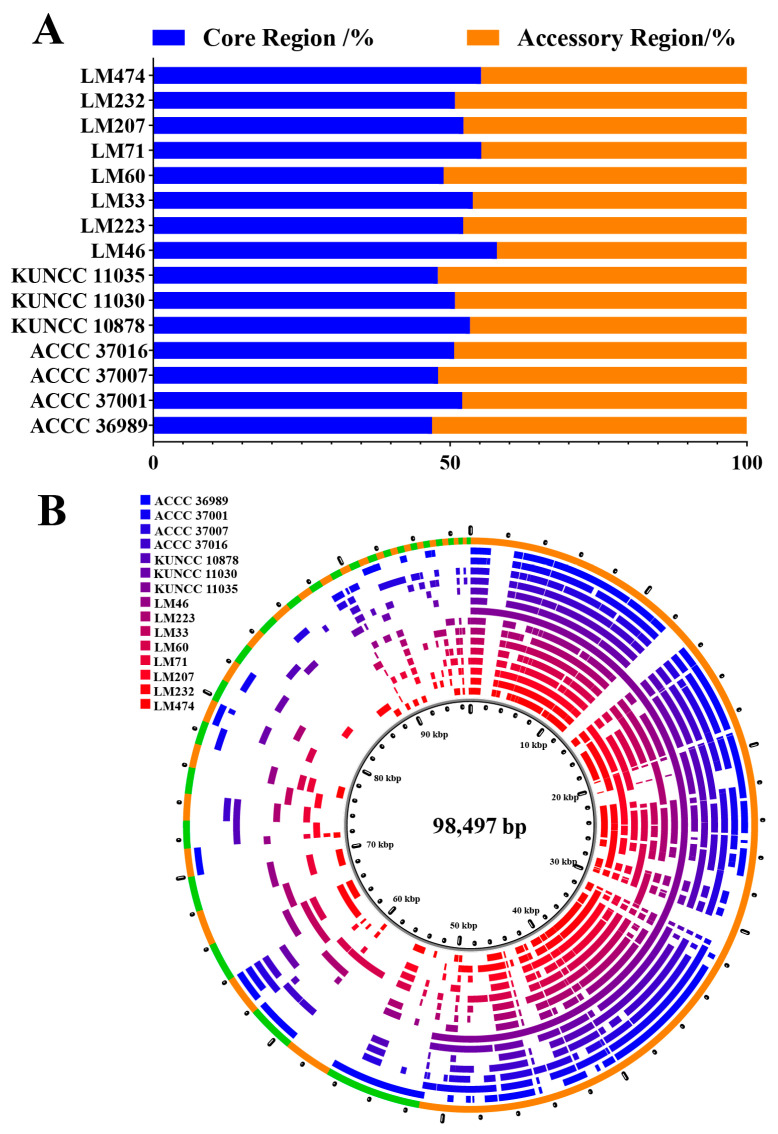
Pan-mitogenome map of *Claviceps purpurea*, depicting core and accessory genomic regions. (**A**) Proportion of core and accessory regions in the mitogenomes of 15 *C*. *purpurea* strains. (**B**) Pan-mitogenome map of 15 *C*. *purpurea* strains.

**Figure 11 biology-14-01548-f011:**
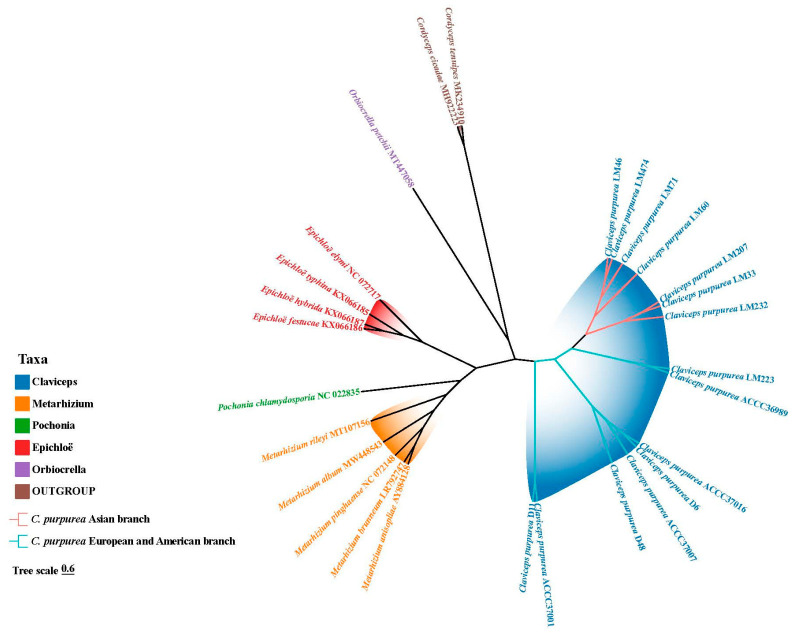
Phylogenetic tree of the family Clavicipitaceae constructed based on the 14 mitogenome protein-coding genes (PCGs).

## Data Availability

The 15 complete mitochondrial genome sequences and their corresponding annotation information involved in this study have been deposited in the GenBank database of the National Center for Biotechnology Information (NCBI) (URL: https://www.ncbi.nlm.nih.gov/genbank/; accessed on 16 October 2024). The accession numbers are PV718106, PV738934, PV658282, PV640608, PV650928, PV658281, PV743142, and PV698189.

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
