# Peer review of "Pan-Mitogenome Construction, Intraspecific Variation, and Adaptive Evolution of the Plant Pathogenic Fungus Claviceps purpurea"

_biology, 2025, doi:10.3390/biology14111548_

Round 1
Reviewer 1 Report
Comments and Suggestions for Authors
I appreciate the comprehensive work presented in this manuscript, particularly the detailed assembly and annotation of 15 Claviceps purpurea mitogenomes and the insightful analysis of intraspecific variation and evolutionary patterns. The study clearly highlights the importance of accessory regions in mitogenome variation, the strong purifying selection on protein-coding genes, and the phylogenetic relationships among Chinese and non-Chinese strains, which adds valuable knowledge to the field.
However, I have identified some scientific errors in the manuscript that need to be addressed. Please review these carefully, make the necessary corrections, and submit the revised version for final consideration.

Author Response
Comments 1: I appreciate the comprehensive work presented in this manuscript, particularly the detailed assembly and annotation of 15 Claviceps purpurea mitogenomes and the insightful analysis of intraspecific variation and evolutionary patterns. The study clearly highlights the importance of accessory regions in mitogenome variation, the strong purifying selection on protein-coding genes, and the phylogenetic relationships among Chinese and non-Chinese strains, which adds valuable knowledge to the field.
However, I have identified some scientific errors in the manuscript that need to be addressed. Please review these carefully, make the necessary corrections, and submit the revised version for final consideration.
Response 1: We sincerely appreciate the time and thoughtful review you have devoted to our manuscript. We are deeply grateful for your positive feedback on our work and the constructive suggestions you have provided. We have carefully examined the scientific errors you identified and made the necessary revisions to the manuscript to ensure the accuracy and rigor of the research. The revised manuscript has been resubmitted, and we would be happy to address any further suggestions or clarifications promptly.
Reviewer 2 Report
Comments and Suggestions for Authors
This study used a rigorous pan-metagenomic analysis of the 15 strains of Claviceps purpurea, a plant fungal pathogen, collected from three different countries in Eastern Asia. This study is significantly important in understanding the key mitochondrial evolutionary adaptation mechanism. The authors have utilized multiple assemblies and annotation tools to ensure data consistency. However, there are a few comments on the manuscript that might strengthen the study more.
- The majority of the isolates (6) were collected from China, which may bias the phylogenetic inference toward East Asian origins.
- The authors have focused mainly on the mitogenomic data; however, the nuclear genome data may strengthen the conclusion about adaptation.
- Any experimental evidence of the significance of the adaptive loss of atp9 might strongly support the hypothesis established by the authors.
Please see the attachment for more comments.

Author Response
Comments 1: The authors used 15 Claviceps purpurea strains from different Geographical sources; however, among those, 7 samples were collected from China, and 8 sequences were retrieved from NCBI, indicating inconsistency in sample collection.
Response 1: We appreciate the reviewers' valuable comments. We believe that combining self-sequenced samples with sequences from public databases was a strategic choice made to maximize the geographic and genetic diversity of our dataset. Relying solely on our seven Chinese samples would have limited the scope of the study. Incorporating sequences from NCBI is a common practice that strengthens our research by providing essential broader context for interpreting the characteristics of Chinese strains. This enables more meaningful comparisons of Claviceps purpurea population structure on a global scale.
Comments2: The authors have performed a detailed computational analysis for mitogenomic assemblies and annotation; however, the brief working principle of these tools is missing. Also, for tRNA secondary structure prediction, if the author can mention the score of the covariance model would be beneficial.
Response2: We sincerely appreciate the reviewers' insightful and constructive suggestions. We fully agree that providing a more detailed methodological description will enhance the clarity and reproducibility of our work. We have revised the manuscript accordingly, with specific modifications as follows.
- Brief description of the tool's working principle:
Regarding the assembly tool's principle, we have added the following to Line 137: “Among these, getorganelle recruits organelle-associated reads using a modified ‘baiting and iterative mapping’ approach, conducts de novo assembly, filters and disentangles the assembly graph, and produces all possible configurations of circular organelle genomes.”
For the annotation tool's working principle, we added the following at Line 144: “MFannot effectively captures evolutionarily conserved feature signals in protein primary sequences and secondary structures by integrating Hidden Markov Models (HMM), Covariance Models (CM), and the ERPIN model.”
- Regarding the addition of covariance model scores for tRNA prediction, we have incorporated this into Table S13, which summarizes all tRNA genes identified within mitochondrial genomes.
Comments3: The author should explain more about the disparity in codon usage by the conserved PCGs between the different strains (lines 208-209).
Resoponse3: We sincerely appreciate the reviewer for raising this important point. We have expanded the codon usage section of the manuscript (Line 239) to provide a detailed explanation of the observed codon usage differences, with the following specific revisions:
“Codon usage analysis revealed a high degree of similarity in codon frequency across strains isolated from different host plants. Overall, codons ending in A/T were used more frequently than those ending in G/T, indicating a strong AT bias. The most frequently used codons were UAA (coding for leucine) and AGA (coding for arginine). The elevated AT content of the mitogenomes (averaging 65.04%) was primarily attributable to the preferential use of codons ending in adenine (A) or thymine (T) (Table S4). Although strains from different host sources exhibited high overall similarity in codon usage frequency (generally favoring A/T-terminating codons), they formed three distinct clusters based on RUSC profiles. This indicates that codon preferences are not entirely uniform among strains, with LM207 showing significant differences in codon usage preferences compared to other strains.”
Comments4: Could the authors explain the nonsynonymous and synonymous substitution rates of the PCGs?
Response4: Thank you very much for your question. I will explain the concepts of non-synonymous substitution rate (Ka) and synonymous substitution rate (Ks):
The concept of KaKs: Due to codon degeneracy, where three nucleotides specify one amino acid, the 64 nucleotide combinations determine 20 amino acids, resulting in redundancy. Typically, a change in the third base results in a synonymous substitution, while changes in the first or second bases result in a non-synonymous substitution. When comparing the sequences of two homologous genes, differences between these segments become apparent. Ultimately, some base changes lead to alterations in the encoded amino acid (non-synonymous substitution), while others do not (synonymous substitution).
Ka/Ks calculation: Ka = number of SNPs with non-synonymous substitutions / number of non-synonymous substitution sites; Ks = number of SNPs with synonymous substitutions / number of synonymous substitution sites.
Relationship between KaKs and evolution: Ka/Ks > 1 indicates positive selection on the gene; Ka/Ks = 1 indicates neutral evolution; Ka/Ks < 1 indicates purifying selection on the gene.
Comments5: The authors showed a higher nucleotide diversity in orf139 and orf143. A brief explanation about the translated products of the mentioned ORFs would be helpful.
Response5: Thank you for your valuable feedback. In our study, we found that the orf139 and orf143 genes exhibit high nucleotide diversity. Previous research indicates that orf139 and orf143 are two common open reading frames in mitochondrial genomes. They are factors associated with cytoplasmic male sterility (CMS) in plant organelle genomes, particularly mitochondria (Wei et al. 2007; Hanson et al. 1999). We have briefly supplemented the functional description of these two ORF genes at line 449 in the original text.
Wei W, Wang H, Liu G. Transcriptional regulation of 10 mitochondrial genes in different tissues of NCa CMS system in Brassica napus L. and their relationship with sterility. J Genet Genomics. 2007 Jan;34(1):72-80. doi: 10.1016/S1673-8527(07)60008-3.
Hanson MR, Wilson RK, Bentolila S, Köhler RH, Chen HC. Mitochondrial gene organization and expression in petunia male fertile and sterile plants. J Hered. 1999 May-Jun;90(3):362-8. doi: 10.1093/jhered/90.3.362.
Comments6: In addition to the above-mentioned comments, I would also like to mention a few points:As the majority of the samples are collected from China, this may raise questions about the phylogenetic inference of Claviceps purpurea towards Asian origin. Along with the mitogenomic data, a detailed nuclear genomic data would have been an additive aspect to the study.
Response6: We appreciate the key questions raised by the esteemed reviewer, which are indeed crucial for accurately interpreting our findings.
- Regarding the geographic origin of samples and phylogenetic inferences:
We wish to clarify that the primary purpose of our phylogenetic analysis was not to definitively determine the geographic origin of C. purpurea, but rather to investigate intraspecific evolutionary relationships and genetic diversity based on mitochondrial genome studies. Although most strains originate from China, our dataset intentionally includes representative strains from other regions (e.g., Europe, North America) as outgroups and comparators. In the manuscript, we state that “Chinese and non-Chinese isolates form distinct lineages,” which reflects an observation of genetic structure rather than an assertion of origin. The observed pattern aligns more with geographic isolation and subsequent diversification than with a specific center of origin.
- Regarding the inclusion of nuclear genome data:
We fully agree with the reviewer that incorporating nuclear genome data would be a valuable addition. Nuclear genomes provide complementary evolutionary insights due to differing genetic patterns and evolutionary rates. In fact, we have prioritized nuclear genome data analysis as a focus for our future research. However, the purpose of this study is to fill a critical knowledge gap—the insufficient understanding of mitochondrial genomic intraspecific diversity and evolutionary characteristics in C. purpurea. To our knowledge, this represents the first comprehensive comparative analysis of mitochondrial genomes across multiple C. purpurea strains. Mitochondrial genomes have been demonstrated to be powerful markers for resolving intraspecific phylogeny and understanding genomic plasticity, a point further supported by our findings regarding structural variation and purifying selection.
Comments 7:Any experimental evidence of the significance of the adaptive loss of atp9 might strongly support the hypothesis established by the authors.
Response 7:Thank you very much for this profound and highly constructive comment. The point you raise is indeed crucial. In this study, we identified a deletion of the atp9 gene encoding a subunit of the ATP synthase complex. However, this phenomenon is not unique; large-scale deletions of the atp9 gene have also been observed in lichen fungi. In that study, the authors suggested this deletion is associated with a unique lifestyle of parasitic fungi. Photosynthetic symbionts can transfer energy molecules to lichen fungi rather than transferring functional atp9 mRNA or proteins between symbionts. This “subfunctionalization” would free the lichen fungal genome from investing in energy production, thereby enabling greater investment in other processes such as reproduction(Pogoda et al., 2018). Therefore, we hypothesize that the atp9 gene deletion in the obligate parasite Claviceps purpurea represents an adaptation for streamlining and optimizing metabolic pathways. We fully recognize its significance and have planned experiments to validate the functional consequences of atp9 gene loss.
Pogoda CS, Keepers KG, Lendemer JC, Kane NC, Tripp EA. Reductions in complexity of mitochondrial genomes in lichen-forming fungi shed light on genome architecture of obligate symbioses. Mol Ecol. 2018, 27:1155-1169. doi: 10.1111/mec.14519.
Reviewer 3 Report
Comments and Suggestions for Authors
Dear authors,
Overall, the paper is well presented. I would only ask if you could double check the numbers of PCGs, because the line 82 claims that 13 were found, while other parts of the text says 14. Please double check the other numbers too. I am also missing an information: how many strains were from China and how many from the UK and Canada? could you provide this information in the beginning of the Materials and Methods section? If there were more strains from China, please mention that in the Results or Discussion section too, as a limitation of the work.
Comments on the Quality of English LanguageOverall, the text is well written and I only made a few suggestions that would help to improve the reading.
L45: Replace „an remarkably“ with „a remarkably“
L47: Replace „With a worldwide distribution“ with „with worldwide distribution“
L91-95: Please correct to „…comprising 15 C. purpurea strains collected from the host plants Roegeria kamoji, Avena sativa (oat) and Bromus riparius in China, Canada and the United Kingdom.“
L101: Replace „This study included 15 C.purpurea strains from host plants belonging to 13 genera, collected in China, Canada, and the United Kingdom.“ With „Fifteen C. purpurea strains from host plants belonging to 13 genera and collected in three distinct regions (China, Canada and the United Kingdom) were evaluated.
The idea here is to avoid repeating „in this study“ too often (as it also appears in L104).
L106: For the NCBI SRA: If possible, you could write: „were sourced from the NCBI Sequence Read Archive (SRA; available online at: (link here), accessed on dd.mm.yyyy).
L116-120 and L126-135: For the online tools, please use the structure suggested above (available online at …, accessed on …).
L132: Replace „To comprehensively evaluate the structure of repeat elements,…“ with „For a comprehensive evaluation of the structure of repeat elements,…“.
L138: Fix the typo on the word „relationships“.
L165-166: Replace „with strain LM46 exhibiting the smallest genome and the strain KUNCC 11035 possessing the largest“ with „with the strains LM46 and KUNCC 11035 exhibiting the smallest and the largest genomes, respectively“. You can reformulate the phrase as you would prefer for a better reading, but „possessing“ feels out of place here
L178: Replace „number“ with „quantity“.
L193: You can write only the abbreviation „ORFs“ here, as the term was already mentioned before.
L239-240: merge the second phrase „It was showed that …“ with the first phrase: „…isolated from different host plants (Figure 5), suggesting that the substitution rates varied across genes.“
Figures 4 and 5: As Figures need to be independent from the text, please write the meaning of the abbreviation „PCGs“.
L282: Replace „may“ with „might“
L325: Place a comma after „processes“
L342: Please fix the beginning of the phrase to „A phylogenetic tree of the family…“
L343: Merge the second phrase („Phylogenetic analysis were conducted…“) with the first one. For example: „… from 13 PCGs, and the phylogenetic analysis was conducted employing both maximum likelihood (ML) and Bayesian inference (BI).“. If the abbreviations ML and BI won’t appear often, consider removing them.
L347: I think the word „respectively“ is not necessary here.
L348: „Claviceps was found to be the closest to…“.
L349: I am confused, did they all clustered with Claviceps or there were five clades as mentioned in L347?
L352: Remove the comma after the word „lineage“. Instead, place it on L353 before the word „suggesting“.
L359: You can also write C. purpurea here.
L363: You can write only the abbreviation „ORFs“ here.
L400: It is already mentioned in the topic above that the analysis was conducted with 15 C. purpurea strains derived from diverse hosts and geographic regions, so I suggest the change oft he phrase to „Our analysis revealed substantital intraspecific variation among the strains studied here (…)“
L413: „…revealed high nucleotide diversity (Pi) regions…“
L415: Place a comma before „such as“
With kind regards
Author Response
Comments 1:Overall, the paper is well presented. I would only ask if you could double check the numbers of PCGs, because the line 82 claims that 13 were found, while other parts of the text says 14. Please double check the other numbers too. I am also missing an information: how many strains were from China and how many from the UK and Canada? could you provide this information in the beginning of the Materials and Methods section? If there were more strains from China, please mention that in the Results or Discussion section too, as a limitation of the work.
Response 1: Thank you very much for your careful review of this manuscript and for your valuable comments. We have verified the number of PCGs and made the corresponding revisions in the manuscript. Additionally, we have added the number of strains from China, the UK, and Canada to the Materials section of the article. We have also included limitations arising from the high proportion of Chinese strains in the materials section.
Comments 2:L45: Replace “an remarkably” with “a remarkably”
Response 2:Thank you very much for your suggestion. We have already replaced “an remarkably” with “a remarkably” at line L45 in the original manuscript.
Comments 3: L47: Replace “With a worldwide distribution” with “with worldwide distribution”
Response 3: Thank you very much for your suggestion. We have already replaced “With a worldwide distribution” with “with worldwide distribution” at line L47 in the original manuscript.
Comments 4: L91-95: Please correct to “…comprising 15 C. purpurea strains collected from the host plants Roegeria kamoji, Avena sativa (oat) and Bromus riparius in China, Canada and the United Kingdom.”
Response 4: We sincerely appreciate the time and effort you have devoted to reviewing our manuscript. We have made the requested modifications at L91-95.
Comments 5: L101: Replace “This study included 15 C. purpurea strains from host plants belonging to 13 genera, collected in China, Canada, and the United Kingdom.” With “Fifteen C. purpurea strains from host plants belonging to 13 genera and collected in three distinct regions (China, Canada and the United Kingdom) were evaluated.
Response 5: We sincerely appreciate the time and effort you have devoted to reviewing our manuscript. We have revised the original manuscript at L101, replacing “This study included 15 C. purpurea strains from host plants belonging to 13 genera, collected in China, Canada, and the United Kingdom” with “Fifteen C. purpurea strains from host plants belonging to 13 genera and collected in three distinct regions (China, Canada, and the United Kingdom).”
Comments 6: The idea here is to avoid repeating “in this study” too often (as it also appears in L104).
Response 6: We sincerely appreciate the time and effort you have devoted to reviewing our manuscript. We have made the revisions to the original manuscript at L104 as per your suggestion.
Comments 7: L106: For the NCBI SRA: If possible, you could write: “were sourced from the NCBI Sequence Read Archive (SRA; available online at: (link here), accessed on dd.mm.yyyy).”
Response 7: We sincerely appreciate the time and effort you have devoted to reviewing our manuscript. We have made the revisions to the original manuscript at L106 as per your suggestion.
Comments 8: L116-120 and L126-135: For the online tools, please use the structure suggested above (available online at …, accessed on …).
Response 8: Thank you very much for your valuable suggestions. We have made the revisions to the original manuscript as requested.
Comments 9: L132: Replace “To comprehensively evaluate the structure of repeat elements,…” with “For a comprehensive evaluation of the structure of repeat elements,…”.
Response 9: Thank you very much for your valuable suggestions. We have replaced “To comprehensively evaluate the structure of repeat elements,…” with “For a comprehensive evaluation of the structure of repeat elements,…” at line 132 of the original text, as per your suggestion.
Comments 10: L138: Fix the typo on the word “relationships”.
Response 10: Thank you very much for pointing this out. We have corrected the spelling error in “relationships” at line 138 of the original manuscript.
Comments 11: L165-166: Replace “with strain LM46 exhibiting the smallest genome and the strain KUNCC 11035 possessing the largest” with “with the strains LM46 and KUNCC 11035 exhibiting the smallest and the largest genomes, respectively”. You can reformulate the phrase as you would prefer for a better reading, but “possessing” feels out of place here
Response 11: Thank you very much for your valuable suggestions. We have revised the text on pages L165-166 from “with strain LM46 exhibiting the smallest genome and the strain KUNCC 11035 possessing the largest” to “with the strains LM46 and KUNCC 11035 exhibiting the smallest and the largest genomes, respectively.”
Comments 12: L178: Replace “number” with “quantity”.
Response 12: We sincerely appreciate your correction and have made the necessary revision at line 178 of the original manuscript.
Comments 13: L193: You can write only the abbreviation “ORFs” here, as the term was already mentioned before.
Response 13: We sincerely appreciate your correction and have made the necessary revision at line 193 of the original manuscript.
Comments 14: L239-240: merge the second phrase “It was showed that …” with the first phrase: “…isolated from different host plants (Figure 5), suggesting that the substitution rates varied across genes.”
Response 14: We sincerely appreciate your suggestions. As requested, we have merged the content at L239-240 of the original manuscript.
Comments 15: L239-240: merge the second phrase “It was showed that …” with the first phrase: “…isolated from different host plants (Figure 5), suggesting that the substitution rates varied across genes.”
Response 15: We sincerely appreciate your suggestions. As requested, we have merged the content at L239-240 of the original manuscript.
Comments 16: Figures 4 and 5: As Figures need to be independent from the text, please write the meaning of the abbreviation “PCGs”
Response 16: We sincerely appreciate your suggestion. We have provided an explanation for the abbreviation “PCGs” appearing in Figures 4 and 5.
Comments 17: L282: Replace “may” with “might”
Response 17: We sincerely appreciate your correction. We have amended the word “may” to “might” at line L282 in the original manuscript.
Comments 18: L325: Place a comma after “processes”.
Response 18: We sincerely appreciate your corrections and have made the necessary revisions at line 325 of the original manuscript.
Comments 19: L342: Please fix the beginning of the phrase to “A phylogenetic tree of the family…”
Response 19: We sincerely appreciate your corrections and have made the necessary revisions at line 342 of the original manuscript.
Comments 20: L343: Merge the second phrase (“Phylogenetic analysis were conducted…”) with the first one. For example: “… from 13 PCGs, and the phylogenetic analysis was conducted employing both maximum likelihood (ML) and Bayesian inference (BI).”. If the abbreviations ML and BI won’t appear often, consider removing them.
Response 20: We sincerely appreciate your suggestions and have made the revisions you recommended at line 343 of the manuscript.
Comments 21: L347: I think the word “respectively” is not necessary here.
Response 21: We sincerely appreciate your suggestion. We have removed “respectively” from line L347 of the manuscript as you recommended.
Comments 22: L348: “Claviceps was found to be the closest to…”.
Response 22: We sincerely appreciate your suggestions and have made the necessary revisions at line L348 in the manuscript.
Comments 23: L349: I am confused, did they all clustered with Claviceps or there were five clades as mentioned in L347?
Response 23: Thank you for your valuable question. It is my honor to address your concerns. In this study, phylogenetic analysis identified 15 Claviceps species within the Claviceps genus, forming five distinct branches within the genus.
Comments 24: L352: Remove the comma after the word “lineage”. Instead, place it on L353 before the word “suggesting”.
Response 24: We sincerely appreciate your corrections and have made the requested changes at line 352 of the original manuscript.
Comments 25: L359: You can also write C. purpurea here.
Response 25: We sincerely appreciate your corrections and have made the requested changes at line 359 of the original manuscript.
Comments 26: L363: You can write only the abbreviation “ORFs” here.
Response 26: We sincerely appreciate your suggestion and have abbreviated the text at line L363 in the manuscript as requested.
Comments 27: L400: It is already mentioned in the topic above that the analysis was conducted with 15 C. purpurea strains derived from diverse hosts and geographic regions, so I suggest the change oft he phrase to “Our analysis revealed substantital intraspecific variation among the strains studied here (…)”
Response 27: We sincerely appreciate your suggestions and have made the requested revisions at line L400 in the manuscript.
Comments 28: L413: “…revealed high nucleotide diversity (Pi) regions…”
Response 28: We sincerely appreciate your suggestions and have made the requested revisions at line L413 in the manuscript.
Comments 29: L415: Place a comma before “such as”
Response 29: We sincerely appreciate your suggestion. We have added a comma before “such as” at line L415 in the manuscript as requested.